# Terrestrial Image-Based Point Clouds for Mapping Near-Ground Vegetation Structure: Potential and Limitations

**Luke Wallace** [1,2,*] **, Bryan Hally** [1,2] **, Samuel Hillman** [1] **, Simon D. Jones** [1,2] **and Karin Reinke** [1,2]

1    School of Science, RMIT University, Melbourne 3000, Australia; bryan.hally@rmit.edu.au (B.H.); samuel.c.hillman@gmail.com (S.H.); simon.jones@rmit.edu.au (S.D.J.); karin.reinke@rmit.edu.au (K.R.)
2    Bushfire and Natural Hazards CRC, East Melbourne 3002, Australia
\*    Correspondence: lukeowallace@yahoo.com.au or luke.wallace2@rmit.edu.au; Tel.: +6-13-9925-9726

**Abstract:** Site-specific information concerning fuel hazard characteristics is needed to support wildfire management interventions and fuel hazard reduction programs. Currently, routine visual assessments provide subjective information, with the resulting estimate of fuel hazard varying due to observer experience and the rigor applied in making assessments. Terrestrial remote sensing techniques have been demonstrated to be capable of capturing quantitative information on the spatial distribution of biomass to inform fuel hazard assessments. This paper explores the use of image-based point clouds generated from imagery captured using a low-cost compact camera for describing the fuel hazard within the surface and near-surface layers. Terrestrial imagery was obtained at three distances for five target plots. Subsets of these images were then processed to determine the effect of varying overlap and distribution of image captures. The majority of the point clouds produced using this image-based technique provide an accurate representation of the 3D structure of the surface and near-surface fuels. Results indicate that high image overlap and pixel size are critical; multi-angle image capture is shown to be crucial in providing a representation of the vertical stratification of fuel. Terrestrial image-based point clouds represent a viable technique for low cost and rapid assessment of fuel structure.

**Keywords:** Structure from Motion; vegetation structure; fuel hazard; Terrestrial Laser Scanning

## 1. Introduction

Site-specific information concerning fuel hazard characteristics is needed to support wildfire management interventions and fuel hazard reduction programs [1]. This information is routinely collected through visual-based assessment using photographic guidelines and hazard scoring systems, such as provided in Gould et al. [1] and Hines et al. [2]. These guidelines have been developed to provide an easy to implement, interpret, and systematize method for assessing potential fire behavior in the field. They aim to account for the vertical and horizontal stratification of fuel load as well as the proportion of dead fuel or fuel moisture content [1]. Nevertheless, it is well known that these methods are subjective, resulting in variation in estimates of fuel hazard due to the observer and the rigor applied in making an assessment [3,4]. As such, improved management of fire prone landscapes requires the development of objective, accurate, and repeatable methods for reliable and quantitative measurements of forest fuel attributes.

The vertical continuity of fuel layers can affect the probability of the transition of fire activity to crown layers. Guidelines for estimating fuel load and distribution often indicate measuring fuel

across horizontal space as well as indicating the presence of fuel in vertical layers (i.e., ladder fuels). Gould et al. [1] for instance, define five fuel layers: surface, near-surface, elevated, intermediate, and canopy layers. While airborne and satellite remote sensing has been shown capable of estimating information in the upper fuel layers, measuring surface and near-surface fuels remains difficult, due to signal obscuration and the measurement precision required to discriminate between the ground and fine fuels occurring in sub-canopy layers [5–7]. As such, terrestrial remote sensing techniques are being engaged to attempt to quantify the attributes of these layers.

Terrestrial laser scanning (TLS) has been demonstrated as a practical source of information describing the 3D properties of the sub-canopy layers [8,9]. Garcia et al. [8] demonstrated the capabilities of TLS data for accurately describing a number of forest properties related to canopy, while Gupta et al. [10] demonstrated that changes to near-surface fuels can be observed with this technology. Liang et al. [11], however, outlined that TLS remains a relatively high cost technology requiring expertise at all stages of its operation. Photography has also been used to support the assessment of the near-surface layer and has also been used to quantify coverage [12], identify broad taxonomic classes [13], determine levels of curing and moisture content or stress [13], and estimate fuel load [14]. In general, however, the methods used in the aforementioned studies utilize individual photos. Such techniques can be considered most useful in communities with simple vertical structures [15].

Recently, advances in 3D computer vision and photogrammetric techniques have allowed for the creation of dense point cloud information. Watson et al. [3], Bright et al. [16], Wallace et al. [17], Hillman et al. [18], Jurado et al. [19], and Cooper et al. [20] have demonstrated that vegetation mapping can be achieved using 3D point clouds generated from terrestrially captured images. In these studies, a variety of image capture techniques are used to capture both descriptions of woody vegetation [21], and near-surface vegetation [16,18,20]. The results from the studies, which focus on fine vegetation elements, demonstrate that a moderate to strong correlation with biomass ($r^2$ between 0.54–0.9 depending on the type of vegetation shown in [17]), percentage cover, and height can be achieved with intensive image capture over small areas. Jurado et al. [19] also demonstrated that these point clouds provide a sufficient representation of the environment for accurate 3D classification of the vegetation elements present in the scene.

The promising results presented in Bright et al. [16], Wallace et al. [17], Hillman et al. [18], Cooper et al. [20] suggest terrestrial imagery may be a viable tool for measuring surface and near-surface fuel properties important for fire risk and behavior analysis. For these purposes, sampling and risk assessment methods that balance repeatably, exhaustiveness, and efficiency are required [22]. Despite the promising work completed thus far, this balance is yet to be demonstrated for the sampling of near-ground vegetation fuel properties using image-based point clouds.

The aim of this paper is to provide an assessment of image-based point clouds in measuring fuel properties in near ground layers. This assessment will include an examination of methodological considerations (ground sampling distance (GSD), the distance between center points of neighbouring pixels—commonly referred to as pixel size, and image count) required to collect a point cloud that accurately and repeatably represents fuel hazard properties. Additional commentary is provided on the time, cost, and expertise required to capture this information.

## 2. Methods

### 2.1. Study Area

This study was carried out in Yarra Bend Park situated east of Melbourne, Australia (S 37.7941°, E 145.0104°). Five 2 × 2 m square plots comprised of varying vegetation structure and densities were examined; these plots were subjectively located to be representative of common fuel landscapes in South East Australia (Figure 1). The plots were selected across different surface vegetation strata which can be broadly described as manicured lawn grass (Plot 1), native tall grass (plains grassy woodland)

(Plot 2), dry sclerophyll (box ironbark) forest (Plots 3 and 4), and low shrubs (riverine escarpment scrub) (Plot 5) (Figure 1). Plot centers were established within an arbitrary coordinate system, defined by coordinating six reference targets (crosses on ranging poles placed on the plot corners) surrounding the plot, using a total station.

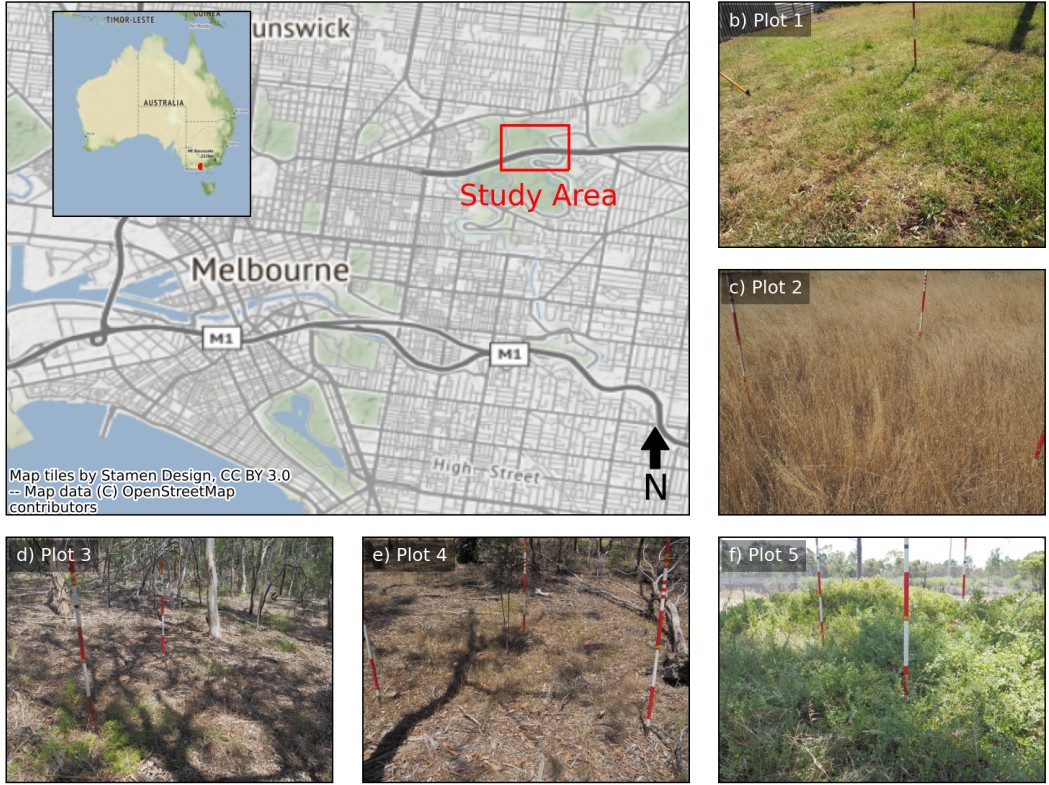

**Figure 1.** The location of the study area (**a**) and the five plots used to demonstrate the utility of image-based point clouds for measuring the 3D properties of near-ground vegetation in (**b**) manicured turf, (**c**) native tall grass, (**d**,**e**) dry schlerophyll forest, and (**f**) low shrub environments.

*2.2. TLS Reference Data*

A Trimble TX8 terrestrial laser scanner (TLS) was used to collect point clouds of each plot. The scanner was set up at four locations surrounding each plot at approximately 5 m from the plot center. For each scan, the scanner was set to record a full hemisphere at a resolution of 11.3 mm at 30 m. The four scans were then registered using cloud to cloud matching within Trimble Realworks 10.1. The co-registered scans were then merged and transformed into the plot coordinate system on the basis of the manually identified locations of the six reference targets.

*2.3. Image Capture*

Images were collected following the TLS surveys with a 10 megapixel resolution Olympus OM-D EM-10 camera with a 14 mm lens. The camera settings were set, depending on the lighting conditions at each plot. Images were acquired in raw format at an approximate horizontal spacing of 20 to 30 cm, following a circular path at a set distance from the plot center (Figure 2). At each exposure point, images were captured sequentially at above ground heights of approximately 0.6, 1.2, and 1.8 m. Images were captured obliquely with the center of the camera's field of view aimed at the plot center. This data capture was repeated three times for each plot at distances of approximately 3, 6, and 9 m from the plot center, resulting in GSD's of 0.08, 0.16, and 0.24 cm and approximately 60, 120, and 180 images being captured, respectively.

### 2.4. Image-Based Point Cloud Generation

Point clouds were generated from the imagery using Agisoft Photoscan Professional software version 1.2.3 (Agisoft LLC, Moscow, Russia). The high quality matching setting was used to generate a sparse point cloud. Scale and co-location was achieved by manually digitizing the location of the six reference marks within 6 to 8 images each. Following this, high quality dense point clouds with mild filtering were generated.

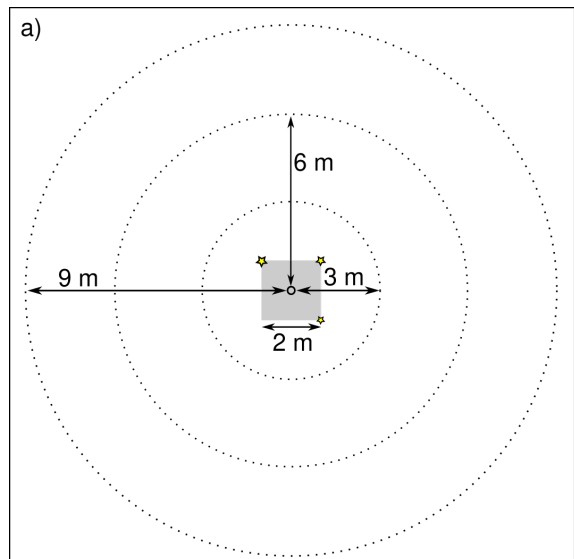
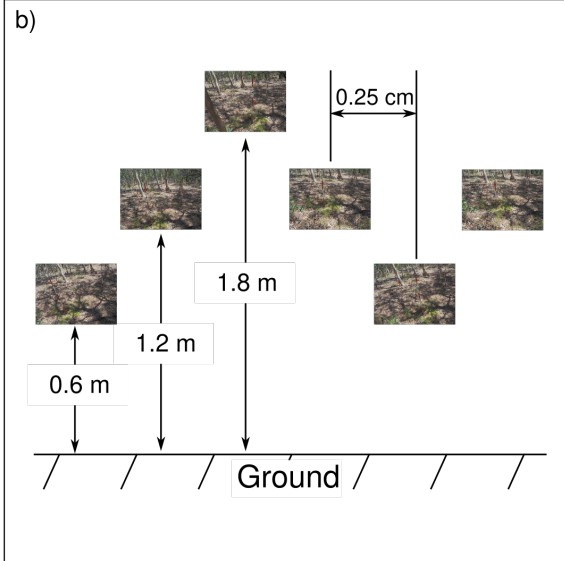

**Figure 2.** The plot layout showing (**a**) horizontal and (**b**) vertical distribution of the image network. Panel (**a**) shows the unobserved plot area in light grey, the imaging locations at all three distances as black dots, and the location of control targets as stars.

In order to test the effect of image capture configurations, point clouds were generated from all photos at each distance and from various sub-samples of the photo sets taken at 3 m. Sub-sampled image sets were generated to assess the effect of a reduction in the number of photos. Sub-sampled image sets consisting of 50% and 75% of all images were generated using a stratified random approach. Images were stratified according to distance along the image path. For 50%, this involved randomly selecting one of two neighboring images. For 75% of the images, the image path was divided into 2 m segments containing 8 images from which 6 random images were selected. This approach was used to ensure sufficient image overlap remained between photos in the final data set. In order to test the repeatability of image-based representations of the vegetation, ten stratified-random subsets of images consisting of 75% of the 3 m image sets were generated for each plot. Both the starting point of the stratification and the random selection of images was varied between selections.

### 2.5. Extraction of Fuel Structure

A variety of metrics can be extracted to assess the quality of representation of vegetation/fuel structure from three-dimensional point clouds derived from a remote sensing source. Herein, we consider the fuel arrangement of the near-ground layers in the horizontal and vertical dimensions separately. To assess the accuracy of the image-based point clouds, a cross-comparison between the metrics obtained from the TLS and the image-based point clouds was undertaken.

Before extracting metrics, each point cloud was clipped to the area of the plot. An absolute height threshold of 2 m above the height at the plot center was applied to remove points originating from the elevated and canopy vegetation layers.

### 2.5.1. Horizontal Coverage

The horizontal coverage of the plot was assessed by examining the percentage of the sampled area observed and point density (or number of points per $cm^2$). These metrics were calculated within a raster with a 1 cm resolution. Increased % area sampled and point densities indicate a higher likelihood that the fine fuel elements present in the landscape are being adequately represented.

### 2.5.2. Vertical Distribution

Assessment of fuel hazard requires the vertical structure or continuity of fuel to be described, as an important driver for fire behavior. Assessing the vertical distribution requires the upper and lower (i.e., the ground) limits to be represented within the 3D point cloud, along with information describing the density of vegetation between these limits. For this study, the 5th and 95th percentile height were used within each 1 cm grid cell to provide an indication of how well information describing the upper and lower limits of vegetation was represented. Percentile heights were calculated on the basis of the height above the minimum TLS height in each grid cell.

### *2.6. Occupied Volume*

To determine how well the 3D distribution of vegetation was represented, each point cloud was projected into a $2\,cm^3$ voxel space. This voxel space was chosen on the basis of the point density achievable from image-based point clouds and the properties of the vegetation being observed. This voxel space provides an indication of vegetation volume through the number of filled voxels. Cross-comparison differences between the voxel space created using the TLS point clouds and the image-based point clouds provide an indication of where information describing the vertical structure is potentially incomplete. To achieve this, and under the assumption that the TLS point cloud as a higher accuracy representation of the true vegetation structure, the Mathews Correlation Coefficient (MCC) and errors of omission and commission were calculated.

To calculate the MCC, the TLS voxel space was considered to contain true vegetation points, as such, any of these voxels in the image-based representation which were empty were considered false negatives. Similarly, any voxels that did not contain points in the TLS representation and did in the image-based representation were considered false positives. The MCC was chosen for this purpose, due to the overwhelming amount of empty space (true negatives) in the voxel space and the fact that MCC displays only high scores if the image based point clouds obtained good results in all of the four categories [23].

## 3. Results

### *3.1. Data Collection, Processing, and Co-Registration*

Image collection at all imaging distances was quicker than the collection of TLS data. Photo capture time ranged from 4 to 12 min, while each TLS scan took 7 min, resulting in a total collection time of approximately of 20–40 min per plot including instrument set-up.

Image-based point clouds were successfully constructed from most photo sets for the plots. However, there were some exceptions. For Plot 1 (manicured turf), two of the ten image sets using 75% of the images failed to produce a point cloud. Similarly, in Plot 5 (low shrubs), the 3 m image set was reduced by 50% and five of the 75% image sets also failed to produce coherent point clouds. The worst performance was observed for Plot 2 (native tall grass), where no image-based point clouds were able to be produced, due to the software being unable to generate reliable feature matches between photos in the long grass environment. The lack of reliable feature matches resulted in no point clouds, or point clouds consisting of only background trees, being created for any combination of the Plot 2 images. The best performance for point cloud construction was observed for Plots 3 and 4.

### 3.2. Point Density and Horizontal Coverage

TLS and image-based point clouds produced high density point clouds at all distances in Plots 1, 3, 4, and 5, Table 1. TLS point clouds had a point density of between 3 and 67 points/cm$^2$. In Plot 2, the TLS point clouds provided minimal plot coverage (57%), due to the obscuration within the surrounding plot. This contributed to the TLS observing this plot at a lower point density. For the image-based point clouds, the point density generally decreased with increasing imaging distance and more so with decreasing photo count (Figure 3). The highest point densities were achieved at Plots 3 and 4 in all image configurations (Table 1). Overall, the MCC demonstrated good agreement (MCC > 0.7) between the TLS and image-based point clouds for Plots 1, 3, and 4, with Plot 5 showing lesser agreement (MCC = 0.57).

**Table 1.** Metrics describing the horizontal and vertical distribution of vegetation elements extracted from the terrestrial laser scanning (TLS) and image-based point clouds. Standard deviation is provided for the 3 m 75% representation based on point clouds derived from the 10 randomly selected photo sets.

| Plot | Technology<br>Distance<br>Image Setup | TLS<br><br>100% | Image-Based | | | | 6 m<br>100% | 9 m<br>100% |
|---|---|---|---|---|---|---|---|---|
| | | | 3 m | | | | | |
| | | | 50% | 75% | 75% SD | | | |
| Plot 1 | PD (points/cm$^2$) | 26 | 26 | 1 | 2 | 0 | 3 | 2 |
| | Coverage (%) | 99 | 94 | 52 | 68 | 9 | 86 | 92 |
| | 5th %ile height (m) | 0.09 | 0.08 | 0.04 | 0.08 | 0.01 | 0.10 | 0.11 |
| | 95th %ile height (m) | 0.22 | 0.18 | 0.17 | 0.19 | 0.05 | 0.23 | 0.22 |
| | Volume (m$^3$) | 0.045 | 0.032 | 0.015 | 0.025 | 0.004 | 0.024 | 0.023 |
| | MCC | | 0.71 | 0.54 | 0.48 | 0.08 | 0.65 | 0.59 |
| | Omission (%) | | 41 | 62 | 65 | 7 | 52 | 64 |
| | Commission (%) | | 13 | 20 | 21 | 7 | 10 | 1 |
| Plot 2 | PD (points/cm$^2$) | 3 | - | - | - | - | - | - |
| | Coverage (%) | 57 | - | - | - | - | - | - |
| | 5th %ile height (m) | 0.04 | - | - | - | - | - | - |
| | 95th %ile height (m) | 0.75 | - | - | - | - | - | - |
| | Volume (m$^3$) | 0.032 | - | - | - | - | - | - |
| | MCC | | - | - | - | - | - | - |
| | Omission (%) | | - | - | - | - | - | - |
| | Commission (%) | | - | - | - | - | - | - |
| Plot 3 | PD (points/cm$^2$) | 67 | 49 | 8 | 9 | 0 | 22 | 1 |
| | Coverage (%) | 100 | 100 | 92 | 94 | 1 | 100 | 60 |
| | 5th %ile height (m) | 0.06 | 0.08 | 0.08 | 0.08 | 0.00 | 0.08 | 0.08 |
| | 95th %ile height (m) | 0.33 | 0.35 | 0.36 | 0.33 | 0.08 | 0.37 | 0.43 |
| | Voluinme (m$^3$) | 0.040 | 0.029 | 0.021 | 0.028 | 0.003 | 0.024 | 0.017 |
| | MCC | | 0.78 | 0.70 | 0.60 | 0.04 | 0.69 | 0.69 |
| | Omission (%) | | 37 | 50 | 54 | 2 | 51 | 49 |
| | Commission (%) | | 4 | 1 | 13 | 3 | 2 | 3 |
| Plot 4 | PD (points/cm$^2$) | 40 | 63 | 10 | 12 | 0 | 12 | 3 |
| | Coverage (%) | 97 | 99 | 93 | 96 | 0 | 96 | 77 |
| | 5th %ile height (m) | 0.13 | 0.13 | 0.12 | 0.13 | 0.00 | 0.12 | 0.13 |
| | 95th %ile height (m) | 0.33 | 0.30 | 0.29 | 0.28 | 0.05 | 0.32 | 0.30 |
| | Volume (m$^3$) | 0.036 | 0.029 | 0.020 | 0.025 | 0.002 | 0.027 | 0.022 |
| | MCC | | 0.73 | 0.70 | 0.67 | 0.01 | 0.70 | 0.69 |
| | Omission (%) | | 39 | 48 | 48 | 2 | 44 | 42 |
| | Commission (%) | | 12 | 4 | 10 | 3 | 11 | 17 |
| Plot 5 | PD (points/cm$^2$) | 48 | 29 | - | 1 | 1 | 1 | 4 |
| | Coverage (%) | 95 | 72 | - | 23 | 15 | 50 | 56 |
| | 5th %ile height (m) | 0.22 | 0.29 | - | 0.27 | 0.19 | 0.44 | 0.45 |
| | 95th %ile height (m) | 0.78 | 0.77 | - | 0.81 | 0.37 | 0.78 | 0.78 |
| | Volume (m$^3$) | 0.085 | 0.039 | - | 0.010 | 0.008 | 0.024 | 0.027 |
| | MCC | | 0.57 | - | 0.11 | 0.10 | 0.47 | 0.46 |
| | Omission (%) | | 58 | - | 94 | 4 | 70 | 74 |
| | Commission (%) | | 22 | - | 55 | 23 | 24 | 18 |

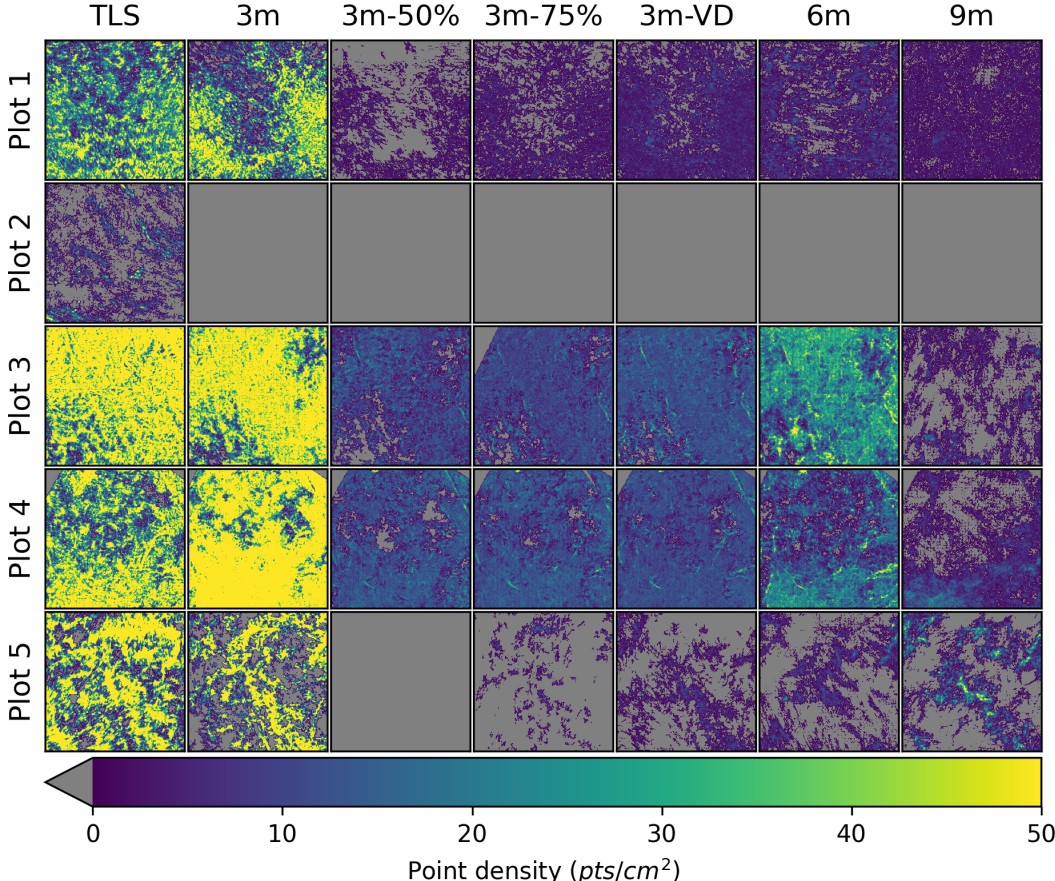

**Figure 3.** Point density and horizontal coverage of each data capture indicated on 1 cm grid for each plot where examples for sub-sampled image combinations were randomly selected. (Plot 1 = manicured turf, Plot 2 = native tall grass, Plots 3 and 5 = dry sclerophyll forest, Plot 5 = low shrubs.).

The horizontal coverage of the image-based point cloud produced using all photos at 3 m provided coverage greater than 90% in all resolved plots except the shrub dominated plot (Plot 5) which had a percentage cover of 72%. Similar to point density, horizontal coverage generally reduced with increasing GSD and decreasing image count. At an imaging distance of 6 m, horizontal coverage remained above 85% in Plots 1, 3, and 4, whilst coverage was reduced by half in Plot 5. At an imaging distance of 9 m, horizontal coverage was further reduced in Plots 3 and 4, however, a slight increase in coverage in comparison to the 6 m point cloud was observed in Plots 1 and 5.

Horizontal coverage was less affected by the reduction of images in Plots 3 and 4. A greater reduction in coverage was seen in Plot 1, where using 75% of the images resulted in a mean decrease of 26% coverage. Horizontal coverage remained similar when using different photosets at 75% of the total image count, with the greatest standard deviations being 9% in Plot 1 and 15% in Plot 5.

*3.3. Vertical Distribution*

The TLS surveys of five sites presented a range of vertical structure with mean vegetation 95th percentile heights ranging from 0.22 m to 0.78 m. Figure 4 shows that the TLS point clouds provide greater penetration in Plots 3 and 5 than any image-based point cloud. Visual inspection of the point clouds confirmed that the penetration provided by the TLS was adequate to provide a good representation of the ground in Plots 1, 3, and 4. The ground was clearly not represented in the TLS point clouds for Plots 2 and 5.

The image-based point clouds captured at 3 m using all images provided a similar representation of the vegetation's vertical structure to the TLS in Plots 1 and 4 (Figure 4). Differences between the

mean TLS and image-based AGH percentiles were minimal in these plots, with the 5th percentile being equal in Plot 4 for the 3 m point cloud. In Plot 1, the 3 m all photos point cloud had a 1 cm lower 5th percentile (Table 1). In Plots 3 and 5, the 5th percentile of the image-based point clouds at 3 m was higher than that found in the TLS representation.

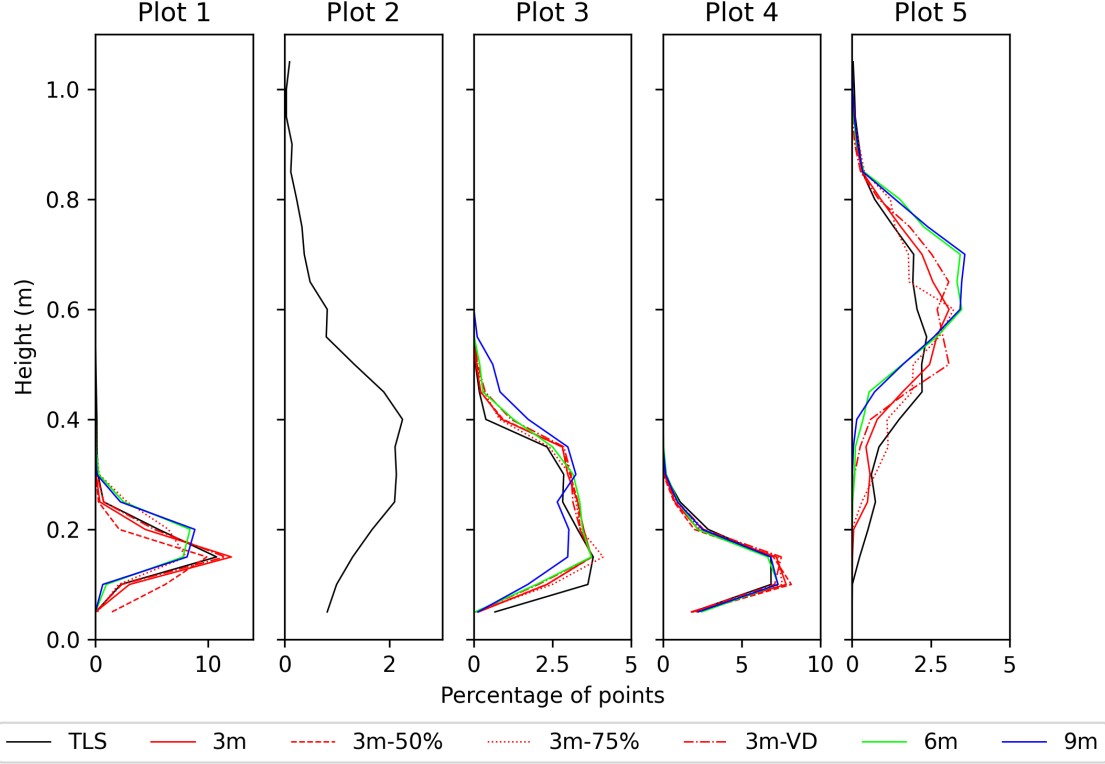

**Figure 4.** Height distributions of the terrestrial laser scanning (TLS) and image-based point clouds captured within the five plots. Distribution plots were created using 5 cm vertical bins.

In all resolved plots except Plot 3, the image-based point cloud captured at 3 m estimated the difference between the 5th and 95th percentile height to be equal to or lower than the TLS estimate, suggesting lower penetration into the vegetation (Table 1). The greatest difference was seen in Plot 5. Analysis of the distribution of difference between the 3 m representation and the TLS representation highlights that the image-based point cloud provides equal or greater penetration in between 20 and 28% of the grid cells Figure 5.

Increasing the GSD resulted in further decreased penetration. At 6 m, the 5th percentile height difference was between −1 cm and 22 cm and between 0 and 23 cm at 9 m. This change in point distribution also resulted in a lower rate of cells having equal or greater penetration than the TLS representation of the environment, with between 7% and 14% and between 6% and 8% when images were captured at 6 and 9 m, respectively. The standard deviation of the 95th percentile height and volume suggest consistent repeatability in the photosets produced using 75% of the imagery in Plots 1, 3, and 4, Table 1. The point clouds did not produce reproducible representations of the environment in Plot 5. In this plot, the standard deviation of volume was 80% of the mean estimate.

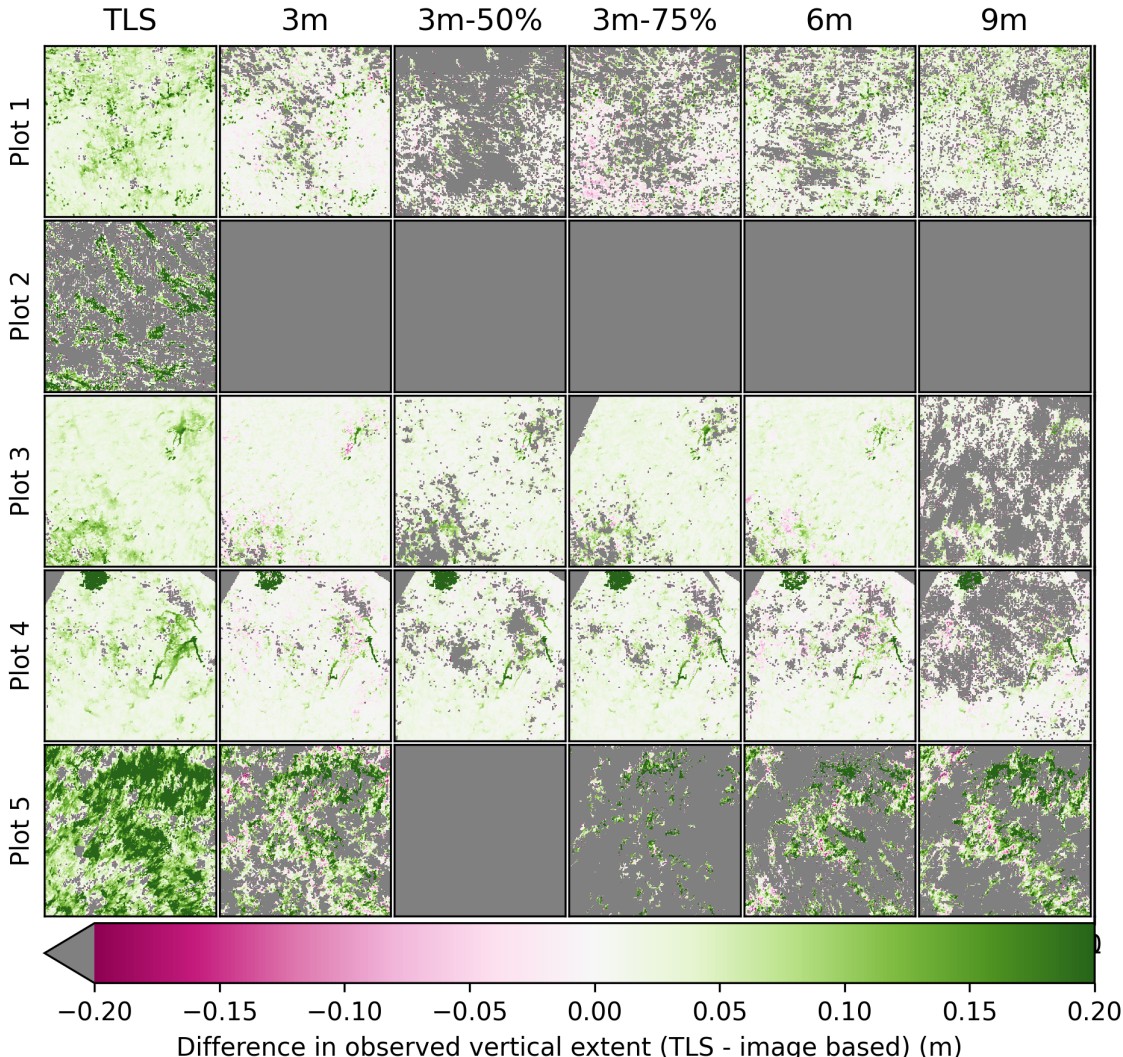

**Figure 5.** Differences between the vertical extent ( 5th–95th percentile) observed between the TLS and each representation produced using the image-based approach.

### 3.4. Occupied Volume

When projected into $2\,cm^3$ voxel space, the volume occupied by the image-based point clouds was less than the volume occupied by the TLS point clouds in all cases. The greatest differences occurred in Plot 5, where the best image-based representation occupied less than 50% of the space occupied by the TLS representation Table 1. In most plots, voxels filled in the TLS voxel space, but not in the space created using the 3 m image-based representation, were caused by fine, often standing, vegetation elements (Figures 6 and 7) such as individual grass-blades. At 6 m, increased omissions occurred in the lower voxel layer of the image-based representation, indicating lower penetration at these distances. At 9 m, the image-based point clouds start to lose fidelity towards the centre of the plots, and the fuel depth and the majority of the fine detail was not represented in the point cloud (Figure 8f for example). Commission in the 3 m image-based point clouds typically occurred in the top-most and bottom-most voxel layers Figure 6. Commission error appeared mostly as noise in the image-based point cloud representation (Figure 8d for example), and due to small misalignments between the representations. In rare instances, however, the image-based technique appears to provide detail not captured by the TLS technology, see Figure 9.

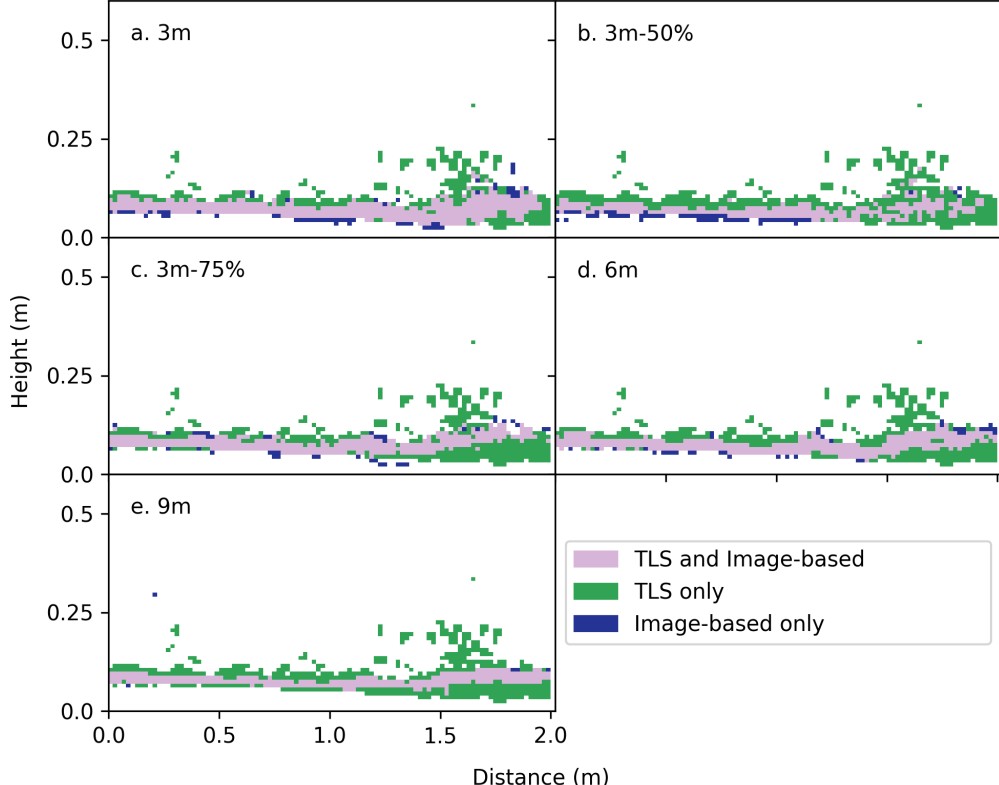

**Figure 6.** Differences in one horizontal row (2 cm wide) between the TLS and image-based voxel representations of the point clouds generated for Plot 1.

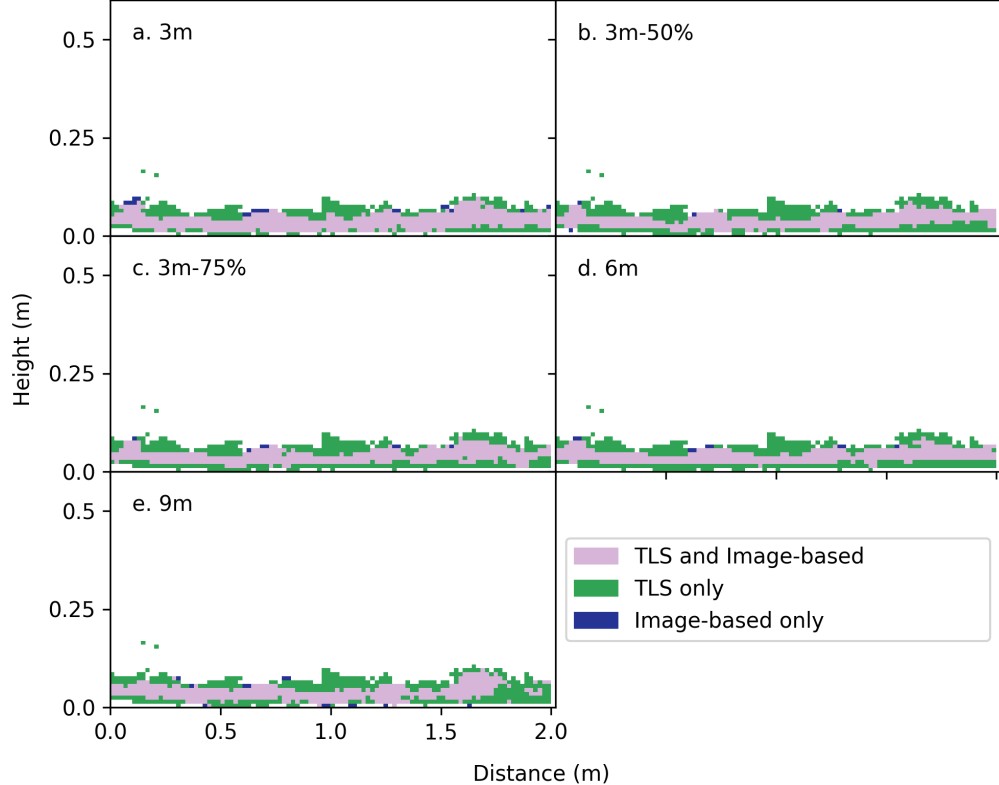

**Figure 7.** Differences in one horizontal row (2 cm wide) between the TLS and image-based voxel representations of the point clouds generated for Plot 3.

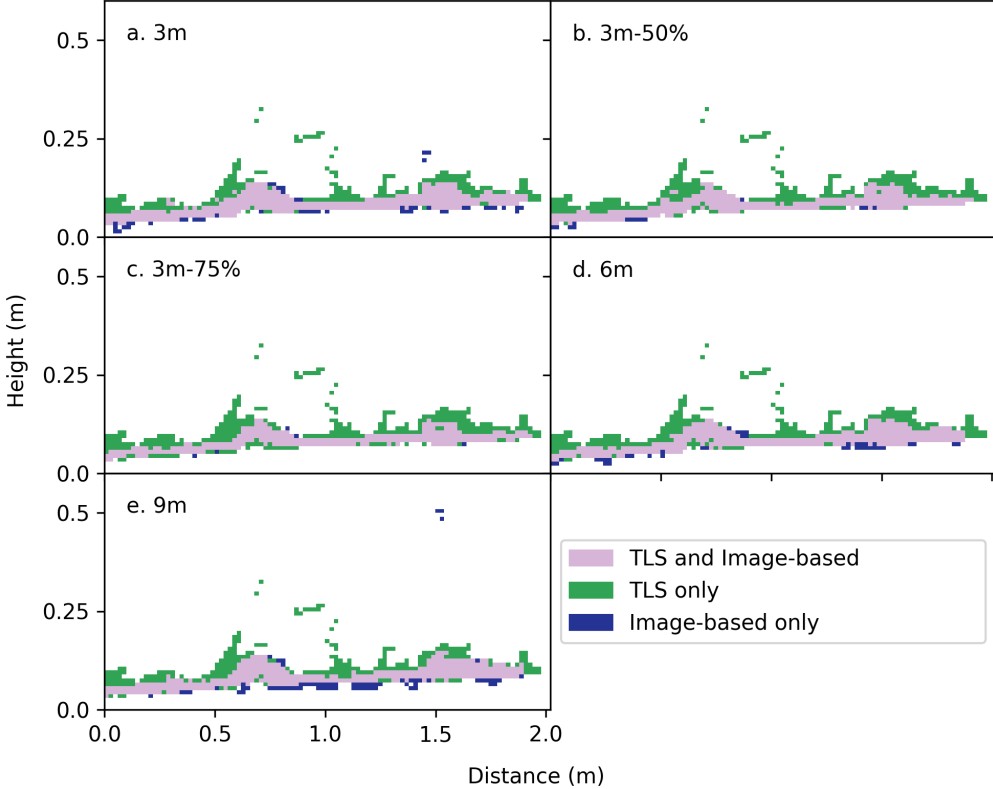

**Figure 8.** Differences in one horizontal row (2 cm wide) between the TLS and image-based voxel representations of the point clouds generated for Plot 4.

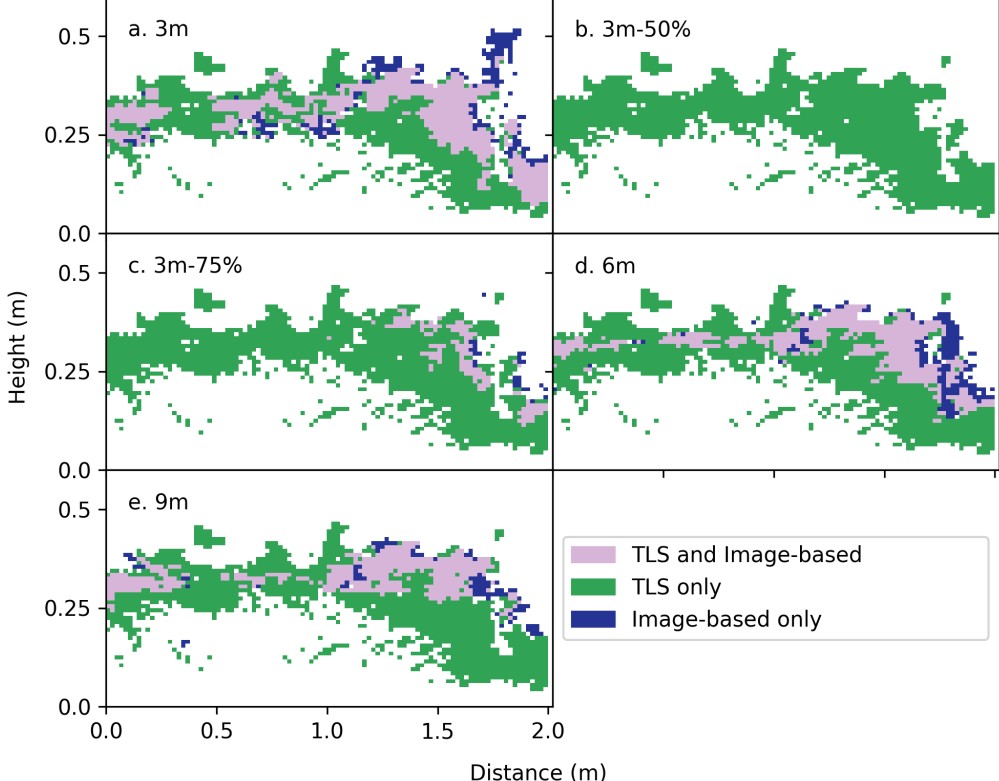

**Figure 9.** Differences in one horizontal row (2 cm wide) between the TLS and image-based voxel representations of the point clouds generated for Plot 5.

## 4. Discussion

Image-based point clouds are being increasingly used for measuring natural environments from airborne [24], unmanned aerial vehicles [25], and hand-held platforms [11,16,18,20]. These studies have shown that in contrast to laser scanning technology, the requirement for advanced equipment and knowledge to collect 3D structural information is reduced [18]. This study contributes to this body of knowledge by examining the utility of image-based point clouds to provide a representation of surface and near-surface fuel hazard.

### 4.1. Image-Based Point Cloud Advantages and Limitations

The method presented herein was developed to require only a consumer grade camera and a source of scale. This represents a significant cost difference to the TLS used in this study with significant difference in price (AUD 800 vs. AUD 200,000). In this study, markers measured using high accuracy surveying techniques were used to define scale. However, this was primarily for the purposes of co-registering the image-based point clouds with the TLS data for cross comparison. Without the need for this level of rigor in co-registration, a device of known and fixed size could be placed in the scene and used to provide scale. This would allow the incorporation of scale without introducing significant error into the final representation of the vegetation, which should reduce the operational requirements of the technique. In comparison to TLS, the infield data collection time was significantly reduced. The processing time to produce a co-registered point cloud and final metrics were similar between the two methods. Notably, the production of an image-based point cloud process required less manual intervention than the TLS registration processing in these environments.

In comparison to commonly used visual field assessments, both remote sensing techniques used in this study provide a quantified description of fuel structure and arrangement. Several studies have demonstrated the accuracy of TLS in describing near-surface vegetation and fuel properties [10,26]. The results from this study highlight that image-based point cloud collection methods are capable of producing a similar representation of the fuel structure to TLS technology in four of the five studied environments. In long grass, the processing software employed did not provide a final point cloud in any image capture configuration. This is likely due to a combination of the fineness of the grass fuels and occlusion caused by the density of vegetation in and surrounding the plot. In this environment only minimal coverage was provided by the TLS (57%).

Both TLS and terrestrial image-based data capture approaches provide only a small area of information per image capture. In this study, we aimed to demonstrate how variations in the image network affect the resolution of a $4\,m^2$ area plot. In order to characterize an area representative of that commonly used in visual field assessment ($314\,m^2$), the limitations of GSD and image count require that multiple image-based samples would need to be collected within the plot area. Approaches such as repeating the image capture method presented here to increase sample size, or utilizing different capture paths could be undertaken to achieve this. Such an approach would have similar requirements to that of the TLS survey where multiple scan locations are required to gather a more complete picture of a plot area [27].

A range of metrics can be used to assess fuel risk, including the horizontal and vertical arrangement of fuels and the overall fuel loading [2,28]. Whilst the acquired point density of the image-based point clouds approach is similar to TLS and sufficient to represent fine fuel vegetation, it provides a point spacing coarser than the achievable, relative measurement precision of the TLS capture approach. As such, horizontal and vertical coverage measures and volume offer greater sensitivity for comparing the abilities of each of the imaging methods for capturing fuel structure and arrangement. These metrics, when derived from point clouds, have been used to develop regression models for the estimation of biomass and fuel load [4,17,20,21]. As such, the differences in the metrics described in this paper are likely to propagate into differences in estimates of biomass and fuel load. Nevertheless, the strength of the correlation between fuel load metrics is dependent on both

the structure described within the point cloud and the methods used to extract the metrics used in the regression.

The image-based point clouds captured at 3 m showed a similar representation of the upper surface of the fuel to TLS (as represented by the 95th percentile height). However, in these point clouds, the 5th percentile height was overestimated by between 0.5 and 6.4 cm. Similarly, low penetration has been found in image-based point clouds of forest captured from UAVs and has been attributed to canopy occlusion [29]. Lower penetration through vegetation and less representation of fine vegetation elements resulted in an lower estimation of volume in comparison to the TLS point clouds in this study (by between 19% and 54 % for Plots 1, 3, 4 and 5). In order to extract estimates of vegetation volume from image-based point clouds, this underestimation needs to be considered, and an alternative representation of the ground may be required.

*4.2. Image Capture Requirements*

Deterioration in point cloud completeness and agreement was shown to occur with decreasing image count and increasing GSD. At a GSD of 0.24 cm (9 m imaging distance in the case of the camera used in this study), significant reduction in the differences between the 95th and 5th percentiles suggests that the image-based point cloud was approximating a smooth surface and was less representative of the actual vertical structure of the vegetation present. In the dry sclerophyll plots, capturing images at GSD of 0.08 cm and 0.16 cm provided similar representations of vegetation structure to TLS-derived representations. The representation of vegetation provided by image-based point clouds was found to be also sensitive to the number of images. Reduction in coverage and structure represented in the reduced photosets highlight the importance of capturing an image network with a well distributed set of images, whilst maintaining sufficient overlap between images. This becomes of increasing importance with increased vegetation complexity.

The results of this study demonstrate that image count/distributions and GSD, governed by photographic distance and camera resolution, are determined by the properties of the environment under investigation. In certain environments, a relatively large area can be captured rapidly. For example, utilizing a 14 megapixel camera and at a distance of 6 m allows an area of approximately 78 m$^2$ (5 m radius circle) to be represented within a single point cloud in leaf litter environments. In other environments such as grass, these restrictions limit the capture area in any one point cloud, potentially offsetting gains in data capture time over TLS. Nevertheless, increases to the represented area might be achieved using alternative image capture strategies, Ref. [21] for example, utilized images around the boundary as well as within a target area to improve coverage.

## 5. Conclusions

Visual fuel hazard assessments are widely used to inform fire land management activities. These assessments are, however, considered to be highly subjective and only provide qualitative information describing the properties of the fuel. The emergence of new remote sensing technologies and techniques provides opportunities to re-explore how fuel hazard is described and assessed. This paper evaluates passive remote sensing techniques as a supplement to the visual assessment of fuel structure. Terrestrial imagery was obtained at varying distances for the experimental plots, and the resultant point clouds and fuel structure metrics compared favorably against those concurrently acquired from Terrestrial Laser Scanning for most of the sites, albeit with an overall lower estimation of fuel volume. In summary, terrestrial image-based point clouds offer a cheap, rapid, repeatable, and quantitative method for characterizing the 3D structure of the near-surface vegetation in certain fuel types. In environments consisting of fine elements such as tall grass, the ability for the solution to generate 3D point clouds was unresolved. The presented technique allows for rapid data capture, however, limitations on the area captured are determined by the properties of the vegetation.

**Author Contributions:** Conceptualization, L.W., B.H., S.D.J., and K.R.; data curation, L.W.; formal analysis, L.W., S.H., and S.D.J.; methodology, L.W. and B.H.; software, L.W., B.H., and S.H.; writing—original draft, L.W. and B.H.; writing—review and editing, L.W., B.H., S.H., S.D.J., and K.R. All authors have read and agreed to the published version of the manuscript.

**Acknowledgments:** Andrew White is acknowledged for both providing editorial advice and his Rocket League skills. The support of the Commonwealth of Australia through the Bushfire and Natural Hazards Cooperative Research Centre is also acknowledged.

**Funding:** This project was funded by the Commonwealth of Australia through the Bushfire and Natural Hazards Cooperative Research Centre.

**Conflicts of Interest:** The authors declare no conflicts of interest.

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
