# Peer review of "Terrestrial Image-Based Point Clouds for Mapping Near-Ground Vegetation Structure: Potential and Limitations"

_fire, doi:10.3390/fire3040059_

Round 1

Reviewer 1 Report

You have done a good job.
However, in the description of the data collection, you could clarify how you do it with the camera, for an outsider reader, it is confusing (for me it is quite confusing).

And I do not know if you could add an economic comparison of the cost of using TLS or conventional camera.

Check the paper, there's some layout flaw.

Author Response

We thank the reviewer for their constructive and helpful comments on our manuscript. In our response below we attempt to address the reviewers concerns as best we can. We feel that this revised manuscript is an improvement on the previous versions. In the below text associate editor comments are in bold and our response is in regular text.

You have done a good job.
However, in the description of the data collection, you could clarify how you do it with the camera, for an outsider reader, it is confusing (for me it is quite confusing).

We have added a diagram (Figure 2) to demonstrate the photo capture method in more detail.

And I do not know if you could add an economic comparison of the cost of using TLS or conventional camera.

The respective sensor prices are outlined at line 225.

Check the paper, there's some layout flaw.

We have thoroughly checked the paper for layout flaws and corrected where required.

Reviewer 2 Report

This paper concerns the mapping of near-ground fuels for wildfire management through the use of 3D Point clouds generated from Terrestrial Laser Scanning (TLS) and overlapping photos from a consumer-grade digital camera. The justification and methods seem sound.

My only concern is the possible effect of camera aperture on the depth of field (DOF) of the point cloud. In section 2.3, the authors note the camera resolution, shutter speed, and ISO, but not the aperture. Aperture is possibly the most important setting in this case, as it controls the DOF of the photo. Aperture is expressed as a ratio, ½, ¼, ………1/22, etc (also referred to as f/2, f/4, f/22 etc). As the numerator increases, (the ratio, or fraction getting smaller) the DOF increases. If a large aperture was used, the distance to subject ratio becomes important, as the depth of field can drastically change between photos. If a small aperture is used (f/11, f/22), then depth of field will not be as much as an issue.

Also, in order to achieve a proper image exposure, ISO, shutter speed, and aperture all much be in agreement. This varies based upon varying lighting conditions. If shutter speed and ISO are kept at a constant setting, that leads one to believe that aperture was used to correct for varying lighting conditions, which would have the effect of changing the depth of field from site to site. E.g., a site that was heavily shaded, or had an overcast day, would need more light to reach the camera sensor, than a site that is brightly lit. To achieve a proper exposure, then 1 of the 3 primary camera settings would need to be adjusted to correct for scenes that are too bright or too dark. It would make much more sense to vary the ISO or shutter speed than the aperture.

If a constant aperture was used as well, then it may be well to state as such. It is possible that at such a wide viewing angle (14mm) that the effect of aperture is minimal, but it is something that should be taken into account.

Please see the following for minor grammatical and formatting edits:

*manuscript to appears to be in a generic MDPI template, and not the Fire template with the Fire logo.

Line: Judging by your citations, i assume that Terrestrial Laser Scanning is the same thing as ground-based LiDAR? If they are the same, perhaps just a acknowledgement in parentheses; if different, briefly state how they might differ. 

Line 49: add “and” before “Cooper et al”

Line 51: make “description of woody vegetation….” plural “descriptions”

Line 53: “depending on the shown in [17]” does not make sense. Is there a word left out here?

Line 64: hyphenate “near ground” to be consistent with the usage in the rest of the paper

Line 106: add a space between “2m” and “above”

Line 128: add a comma after “represented”

Line 146: personal observation: lack of point clouds in tall grass is likely attributed to sway of the grass between photos. Shrubs are much more stationary.

This paper was very interesting, and I enjoyed reading it.

Author Response

We thank the reviewer for their constructive and helpful comments on our manuscript. In our response below we attempt to address the reviewers concerns as best we can. We feel that this revised manuscript is an improvement on the previous versions. In the below text associate editor comments are in bold and our response is in regular text.

This paper concerns the mapping of near-ground fuels for wildfire management through the use of 3D Point clouds generated from Terrestrial Laser Scanning (TLS) and overlapping photos from a consumer-grade digital camera. The justification and methods seem sound. 

My only concern is the possible effect of camera aperture on the depth of field (DOF) of the point cloud. In section 2.3, the authors note the camera resolution, shutter speed, and ISO, but not the aperture. Aperture is possibly the most important setting in this case, as it controls the DOF of the photo. Aperture is expressed as a ratio, ½, ¼, ………1/22, etc (also referred to as f/2, f/4, f/22 etc). As the numerator increases, (the ratio, or fraction getting smaller) the DOF increases. If a large aperture was used, the distance to subject ratio becomes important, as the depth of field can drastically change between photos. If a small aperture is used (f/11, f/22), then depth of field will not be as much as an issue.

Also, in order to achieve a proper image exposure, ISO, shutter speed, and aperture all much be in agreement. This varies based upon varying lighting conditions. If shutter speed and ISO are kept at a constant setting, that leads one to believe that aperture was used to correct for varying lighting conditions, which would have the effect of changing the depth of field from site to site. E.g., a site that was heavily shaded, or had an overcast day, would need more light to reach the camera sensor, than a site that is brightly lit. To achieve a proper exposure, then 1 of the 3 primary camera settings would need to be adjusted to correct for scenes that are too bright or too dark. It would make much more sense to vary the ISO or shutter speed than the aperture.

If a constant aperture was used as well, then it may be well to state as such. It is possible that at such a wide viewing angle (14mm) that the effect of aperture is minimal, but it is something that should be taken into account.

The reviewer is correct and has spotted an error in the text. We did allow the aperture to vary to achieve correct exposure, however, changed the ISO and exposure at each plot such that the aperture remained small (ranging from f/6.3 through f/11 across all images). As the photos were collected in a short period within each plot no or no significant aperture changes were found within the image sets of each plot. We have adjusted the text to describe the correct camera settings.

Please see the following for minor grammatical and formatting edits:

*manuscript to appears to be in a generic MDPI template, and not the Fire template with the Fire logo.

Line: Judging by your citations, i assume that Terrestrial Laser Scanning is the same thing as ground-based LiDAR? If they are the same, perhaps just a acknowledgement in parentheses; if different, briefly state how they might differ.

Thanks for your comment. We have used the term Terrestrial Laser Scanner as it is the typical terminology used for the technology in studies of this type – whilst ground-based LiDAR may be a functionally accurate term, it is less used.

Line 49: add “and” before “Cooper et al”

Corrected

Line 51: make “description of woody vegetation….” plural “descriptions”

Corrected

Line 53: “depending on the shown in [17]” does not make sense. Is there a word left out here?

Yes we have added type of vegetation”. The paper compares pasture, dry grassy forest and lowland forest

Line 64: hyphenate “near ground” to be consistent with the usage in the rest of the paper

Corrected

Line 106: add a space between “2m” and “above”

Corrected

Line 128: add a comma after “represented”

Corrected

Line 146: personal observation: lack of point clouds in tall grass is likely attributed to sway of the grass between photos. Shrubs are much more stationary.

This is an interesting observation and we have a forthcoming publication looking at the effect of wind on TLS estimates. Wind also likely plays a role in the reconstruction of vegetation from images. However, the day of data collection was selected as it had minimal if any wind. The lack of point clouds in grass is likely attributable to the fine nature of the vegetation type minimising the coherency of feature detection and subsequent matching.

This paper was very interesting, and I enjoyed reading i

Thank you

Reviewer 3 Report

The paper presents a new method for mapping near ground fuel based on imagery. The study compares some metrics in five different quadrats obtained with the « light » technique based on image based points cloud (with a camera) to some references obtained with a Terrestrial LiDAR Scanner used for reference. The paper is generally well written and address a relevant question as the objectivity of visual techniques is often problematic. We think the manuscript is acceptable, but can be improved with some minors revisions.

The main metric of interest in terms of surface fuels is the fuel load (and/or the bulk density in 3D), which are not measured or mapped with this technique (neither with the visual one indeed). This should be mentioned somewhere, as the direct usage of such mapping is still far to be straightforward for fire risk in practice. To our mind, the critical research need is how to use such point clouds to measure fuel quantity, not just structure. The limited penetration of imagery when compared to TLS is especially important in this context as quantitative determination of fuel amount can be estimated from point cloud (Beer-Lambert or Maximum Likelihood Estimator). This would be useful to discuss these elements regarding applications of such point clouds (in section 4) and to add a few references on these topics.

The selected fuels seem pretty light and are poorly described (section 2.1). It would be useful to have information regarding the fuel loads of each quadrat for potential further applications of the method. If not available you might refer to other publications. It would be good to discuss the typical limitations of these methods in case of denser surface fuels (the ground is often not even visible in many fire prone shrublands…).

Ln 10:  this sentence is true but it would be more accurate to link it to the next one. Something like « even if the sensitivity to image height and pixel size were high »

Ln 70 : define GSD here or in the method section, as fire is not a remote sensing journal.

Ln 92 and next: the number of total images per quadrat is missing. This would help to undertand the experimental design

ln 102 and Ln 121 : remove these comments

ln 138 : please be more accurate, because it seems that you have two different metrics here : point density (per cm2) and plot coverage (percent of cm2 that are sampled in a raster grid?), which are not correctly described.

Ln 141 : occupied volume should be described a specific subsection (as coverage and vertical distribution)

ln 152 : based

Ln 319 : half of the conclusion is just a reminder of the research question. Could you develop a little more the results and the perspectives?

Author Response

We thank the reviewer for their constructive and helpful comments on our manuscript. In our response below we attempt to address the reviewers concerns as best we can. We feel that this revised manuscript is an improvement on the previous versions. In the below text associate editor comments are in bold and our response is in regular text.

The paper presents a new method for mapping near ground fuel based on imagery. The study compares some metrics in five different quadrats obtained with the « light » technique based on image based points cloud (with a camera) to some references obtained with a Terrestrial LiDAR Scanner used for reference. The paper is generally well written and address a relevant question as the objectivity of visual techniques is often problematic. We think the manuscript is acceptable, but can be improved with some minors revisions. 

The main metric of interest in terms of surface fuels is the fuel load (and/or the bulk density in 3D), which are not measured or mapped with this technique (neither with the visual one indeed). This should be mentioned somewhere, as the direct usage of such mapping is still far to be straightforward for fire risk in practice. To our mind, the critical research need is how to use such point clouds to measure fuel quantity, not just structure. The limited penetration of imagery when compared to TLS is especially important in this context as quantitative determination of fuel amount can be estimated from point cloud (Beer-Lambert or Maximum Likelihood Estimator). This would be useful to discuss these elements regarding applications of such point clouds (in section 4) and to add a few references on these topics. 

We acknowledge the reviewers concern with regards to the capacity of 3D point clouds to measure fuel load. Nevertheless, whilst current research suggests that fuel load is an important consideration for fuel  management, the way fuel is arranged over the vertical profile is also of significance in its own right (see  Duff et al. 2017 for a good discussion around this topic). 

To emphasise this, we have added the following passage to the manuscripts discussion, 

 A range of metrics can be used to assess fuel risk, including the horizontal and vertical arrangement of fuels and the overall fuel loading \citep{duff2017revisiting, hines2010}. Whilst the acquired point density of the the image-based point clouds approach is similar to TLS and sufficient to represent fine fuel vegetation, it provides a point spacing coarser than the achievable, relative measurement precision of the TLS capture approach. As such, horizontal and vertical coverage measures and volume offer greater sensitivity for comparing the abilities of each of the imaging methods for capturing fuel structure and arrangement. These metrics, when derived from point clouds have been used to develop regression models for the estimation of biomass and fuel load \citep{ wallace2017non, cooper2017examination,spits2017investigating, Liang2015}. As such the differences in the metrics described in this paper are likely to propagate into differences in estimates of biomass and fuel load. Nevertheless, the strength of the correlation between fuel load metrics is dependent on both the structure described within the point cloud and the methods used to extract the metrics used in the regression. 

Reference 

Duff, T. J., Keane, R. E., Penman, T. D., & Tolhurst, K. G. (2017). Revisiting wildland fire fuel quantification methods: the challenge of understanding a dynamic, biotic entity. Forests, 8(9), 351. 

The selected fuels seem pretty light and are poorly described (section 2.1). It would be useful to have information regarding the fuel loads of each quadrat for potential further applications of the method. If not available you might refer to other publications. It would be good to discuss the typical limitations of these methods in case of denser surface fuels (the ground is often not even visible in many fire prone shrublands…). 

Unfortunately, our study area was in a state park (protected ecosystem) precluding the collection of biomass for fuel load measurements. In two of the plots (plot 2 and 5) the ground was not visible, whilst only small patches were visible in the other plots. The technical descriptions of the fuel loads in selected areas have been expanded as requested. 

Ln 10:  this sentence is true but it would be more accurate to link it to the next one. Something like « even if the sensitivity to image height and pixel size were high » 

Given the word limit placed on the abstract we have been unable to incorporate the suggested change. We feel the current text sufficiently addresses the reviewers point. 

Ln 70 : define GSD here or in the method section, as fire is not a remote sensing journal. 

Added definition   "the ground sample distance, the distance between center points of neighbouring pixels -commonly referred to as pixel size 

Ln 92 and next: the number of total images per quadrat is missing. This would help to undertand the experimental design 

We have added the following to the text 

and approximately 60, 120 and 180 images being captured  respectively. 

ln 102 and Ln 121 : remove these comments 

Removed 

ln 138 : please be more accurate, because it seems that you have two different metrics here : point density (per cm2) and plot coverage (percent of cm2 that are sampled in a raster grid?), which are not correctly described. 

Thank you for this comment, we have revised the manuscript to ensure these two metrics are not confused  

Ln 141 : occupied volume should be described a specific subsection (as coverage and vertical distribution) 

 Thank you for this comment, we have revised the manuscript creating a specific subsection 

ln 152 : based 

Correct 

Ln 319 : half of the conclusion is just a reminder of the research question. Could you develop a little more the results and the perspectives? 

We have revised the conclusion as follows

Visual fuel hazard assessments are widely used inform fire land management activities. These assessments are, however, considered to be highly subjective and only provide qualitative information describing the properties of the fuel. The emergence of new remote sensing technologies and techniques provides opportunities to re-explore how fuel hazard is described and assessed. This paper evaluates passive remote sensing techniques as a supplement to the visual assessment of fuel structure. Terrestrial imagery was obtained at varying distances for the experimental plots, and the resultant point clouds and fuel structure metrics comparing favourably against those concurrently acquired from Terrestrial Laser Scanning for the most of the sites, albeit with an overall lower estimation of fuel volume. In summary, terrestrial Image-based point clouds offer a cheap, rapid, repeatable, and quantitative method for characterizing the 3D structure of the near-surface vegetation in certain fuel types. In environments consisting of fine elements such as tall grass, the ability for the solution to generate 3D point clouds was unresolved. The presented technique allows for rapid data capture, however, limitations on the area captured are determined by the properties of the vegetation.

Reviewer 4 Report

General:

This article seems to have been revised satisfactorily, but in two cases, the revisions read like they were copied and pasted from the responses to reviewer comments document. Some additional minor revision is needed before this can be accepted for publication.

Major concerns:

Fig. 1. Indicate in the photos or the caption which plots (1-5) these picture depict! Also, rearrange so that element a) (the map) is in the upper left, and the photos are arranged clockwise after a), to end in the lower left corner.

L102-105. This revised text looks like it may belong in the responses document and not the manuscript. It needs to be rewritten.   

L120-123. Same problem as commented above for L102-105.

Fig. 6-9. These figures would occupy the space on a page more efficiently if the single graph in the bottom row was moved to the left, and the legend moved up to the larger white space thus created to its right.

Minor editorial comments:

L39. The word accurately is inaccurately spelled.

L115. Change “into” to “to”.

L152. Change “density of point” to “point density”.

L159. The rest of the section is written in past tense, so change “are” to “were”.

L165. Delete “is”.

Author Response

We thank the reviewer for their constructive and helpful comments on our manuscript. In our response below we attempt to address the reviewers concerns as best we can. We feel that this revised manuscript is an improvement on the previous versions. In the below text associate editor comments are in bold and our response is in regular text.

General: 

This article seems to have been revised satisfactorily, but in two cases, the revisions read like they were copied and pasted from the responses to reviewer comments document. Some additional minor revision is needed before this can be accepted for publication. 

Major concerns: 

Fig. 1. Indicate in the photos or the caption which plots (1-5) these picture depict! Also, rearrange so that element a) (the map) is in the upper left, and the photos are arranged clockwise after a), to end in the lower left corner. 

Figure 1 has been updated as suggested 

L102-105. This revised text looks like it may belong in the responses document and not the manuscript. It needs to be rewritten. 

Removed    

L120-123. Same problem as commented above for L102-105. 

Removed 

Fig. 6-9. These figures would occupy the space on a page more efficiently if the single graph in the bottom row was moved to the left, and the legend moved up to the larger white space thus created to its right. 

These figures have been updated as suggested. 

Minor editorial comments: 

L39. The word accurately is inaccurately spelled. 

Corrected 

L115. Change “into” to “to”. 

Changed

L152. Change “density of point” to “point density”. 

Changed

L159. The rest of the section is written in past tense, so change “are” to “were”. 

Ok  

L165. Delete “is”. 

Ok